# Double NPY motifs at the N-terminus of the yeast t-SNARE Sso2 synergistically bind Sec3 to promote membrane fusion

**Maximilian Peer[1], Hua Yuan[2†], Yubo Zhang[1†], Katharina Korbula[1], Peter Novick[2]\*, Gang Dong[1]\***

[1]Max Perutz Labs, Vienna Biocenter, Medical University of Vienna, Vienna, Austria; [2]Department of Cellular and Molecular Medicine, University of California, San Diego, La Jolla, United States

**Abstract** Exocytosis is an active vesicle trafficking process by which eukaryotes secrete materials to the extracellular environment and insert membrane proteins into the plasma membrane. The final step of exocytosis in yeast involves the assembly of two t-SNAREs, Sso1/2 and Sec9, with the v-SNARE, Snc1/2, on secretory vesicles. The rate-limiting step in this process is the formation of a binary complex of the two t-SNAREs. Despite a previous report of acceleration of binary complex assembly by Sec3, it remains unknown how Sso2 is efficiently recruited to the vesicle-docking site marked by Sec3. Here, we report a crystal structure of the pleckstrin homology (PH) domain of Sec3 in complex with a nearly full-length version of Sso2 lacking only its C-terminal transmembrane helix. The structure shows a previously uncharacterized binding site for Sec3 at the N-terminus of Sso2, consisting of two highly conserved triple residue motifs (NPY: Asn-Pro-Tyr). We further reveal that the two NPY motifs bind Sec3 synergistically, which together with the previously reported binding interface constitute dual-site interactions between Sso2 and Sec3 to drive the fusion of secretory vesicles at target sites on the plasma membrane.

**\*For correspondence:**
pnovick@ucsd.edu (PN);
gang.dong@meduniwien.ac.at
(GD)

[†]These authors contributed equally to this work

**Competing interest:** The authors declare that no competing interests exist.

## Editor's evaluation

This paper describes a mechanistic and structural investigation of a crucial step in exocytosis that involves the assembly of two t-SNAREs with a v-SNARE. The authors solve the structure of Sec3, a component of the exocyst vesicle tethering complex and the t-SNARE Sso2, and identify a previously unknown binding site that helps to explain the mechanism of this rate-limiting step. The results are of great interest to scientists studying exocytosis, membrane fusion, and protein-protein interaction.

## Introduction

The cytoplasm in eukaryotic cells is compartmentalized into distinct membrane bound organelles. Inter-organelle material exchange is carried out primarily through membrane traffic in which membrane bound transport vesicles bud from a donor compartment and are delivered to a specific acceptor compartment. Upon arriving at the destination, cargo-packed vesicles are first recognized and caught by tethering factors situated on the target membrane, which then hand the captured vesicles over to the soluble *N*-ethylmaleimide-sensitive factor-attachment protein receptor (SNARE) proteins that drive membrane fusion (*Ungermann and Kümmel, 2019*; *Chia and Gleeson, 2014*; *Bröcker et al., 2010*; *Yu and Hughson, 2010*).

There are several types of SNARE proteins, one of which is attached to the membrane of vesicles (v-SNARE), and the others are on the target membrane (t-SNARE). In neuronal exocytosis, fusion of synaptic vesicles to the presynaptic plasma membrane is driven by the assembly of a four-helix bundle containing two t-SNAREs, syntaxin-1 and SNAP-25, on the target membrane and the v-SNARE, synaptobrevin, on synaptic vesicles (*Fernandez et al., 1998*; *Sutton et al., 1998*). Extensive studies have established that SNARE assembly is tightly regulated by multiple auxiliary proteins, including the Sec1/Munc18 (SM) family of proteins, tethering factors, and small GTPases (*Dulubova et al., 1999*; *Fiebig et al., 1999*; *Demircioglu et al., 2014*; *Takemoto et al., 2018*; *Koike and Jahn, 2019*; *Cai et al., 2007*). Munc18 is a chaperone protein that maintains syntaxin-1 in an activated conformation and passes it to its cognate SNARE partners for assembly to catalyze membrane fusion (*Wang et al., 2019*; *Yu et al., 2018*; *Jiao et al., 2018*; *Xu et al., 2010*; *André et al., 2020*).

The exocytic vesicle-docking site in yeast is marked by the octameric exocyst complex, which belongs to the CATCHR family of multi-subunit tethering proteins (*Pleskot et al., 2015*; *Guo et al., 1999*; *Boehm and Field, 2019*). The main function of the exocyst is to capture secretory vesicles at sites of cell surface growth, which include the tip of the daughter cell early in the cell cycle, and the mother-daughter cell junction late in the cycle (*Lipschutz and Mostov, 2002*). The two t-SNAREs for exocytosis in yeast are Sso1/2 and Sec9, which are homologs of syntaxin-1 and SNAP-25, respectively. The v-SNARE attached to secretory vesicles in yeast is Snc1/2, which is equivalent to synaptobrevin in neuronal exocytosis.

Our previous work showed that one of the exocyst components, Sec3, promotes SNARE assembly by interacting with the t-SNARE Sso2 (*Yue et al., 2017*). Sec3 consists of an N-terminal pleckstrin homology (PH) domain, a central putative coiled coil, and a C-terminal helical domain. Like syntaxin-1 and other related t-SNAREs, Sso2 consists of four helices, with the first three (Habc) forming an inhibitory domain and the last (H3) serving as the SNARE motif that interacts with the other two SNAREs during membrane fusion. We have shown that the Sec3 PH domain binds to the auto-inhibited four-helix bundle of Sso2 and promotes a conformational change of the linker between Hc and H3 of Sso2 via an allosteric effect (*Yue et al., 2017*). This change promotes the release of the SNARE motif (H3) of Sso2 and substantially accelerates the formation of the initial binary complex between H3 of Sso2 and the two helices of the other t-SNARE, Sec9. However, it remains unclear how Sso2 is initially recruited to the vesicle target sites marked by the exocyst to drive the efficient fusion reaction between secretory vesicles and the plasma membrane.

Here, we report our structural studies of Sec3-PH in complex with a nearly full-length construct of Sso2 (aa1–270), which lacks only its C-terminal transmembrane region (aa271–295). Our crystal structure of this Sec3/Sso2 complex reveals a previously unknown binding site for Sec3 on Sso2 in addition to the one on its four-helix bundle as reported in our previous work (*Yue et al., 2017*). This extra binding site is located at the N-terminal end of Sso2 and consists of two highly conserved NPY (i.e. Asn-Pro-Tyr) motifs. These NPY motifs are connected to the helical core of Sso2 (i.e. Habc and H3) via a long variable linker. In the two heterodimeric complexes present in the crystal structure, the two NPY motifs of the two Sso2 molecules bind individually to a similar conserved hydrophobic pocket on the two Sec3 molecules. Interestingly, however, our in vitro interaction studies using synthetic polypeptides and recombinant Sec3-PH protein demonstrated that each NPY motif alone bound Sec3 much more weakly than the two NPY motifs together. The importance of the interaction between the NPY motifs of Sso2 and Sec3 was confirmed by a series of in vivo assays in yeast.

Overall, our work has uncovered a new interaction interface and thus establishes dual-site interactions between Sec3 and Sso2, which also suggests potentially an extra regulatory step in exocytic membrane fusion. Binding of the NPY motifs of Sso2 allows efficient recruitment of the t-SNARE protein to the vesicle-docking site on the plasma membrane to facilitate vesicle fusion.

## Results

### Crystal structure reveals two NPY motifs at the N-terminus of Sso2 bound individually to the Sec3 PH domain

We previously reported the structure of Sso2-HabcH3 (aa36–227) in complex with the PH domain of Sec3 (aa75–320) (*Yue et al., 2017*). Recently, we crystallized another complex of the two proteins using a longer version of Sso2 (aa1–270), which contains all Sso2 sequence except for its C-terminal

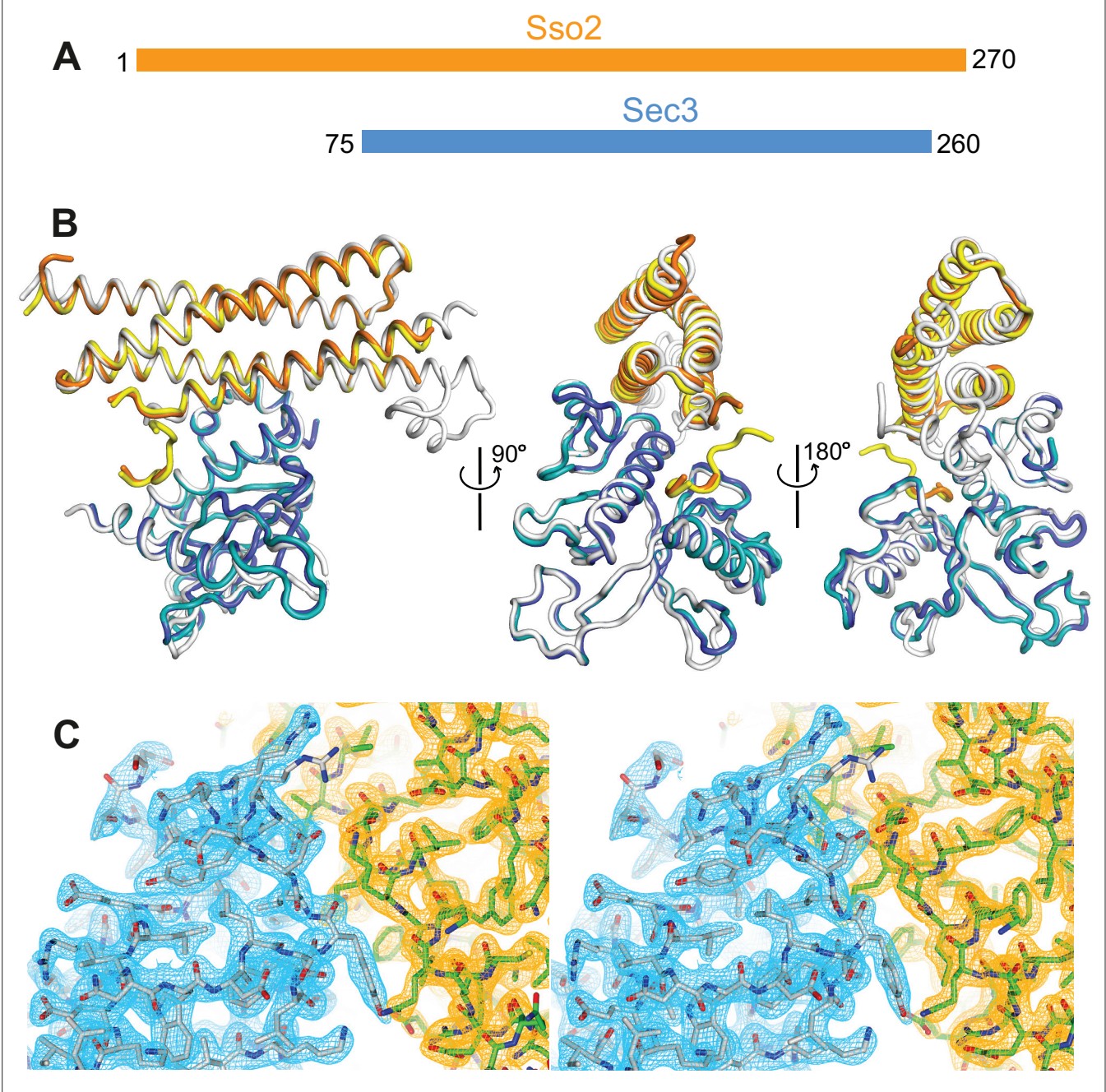

**Figure 1.** Crystal structure of the Sso2/Sec3 complex. (**A**) Schematics of the constructs of Sso2 (aa1–270) and Sec3 (aa75–260) used in our structural studies. (**B**) Superposition of the two Sso2/Sec3 complexes in the asymmetric unit of the crystal structure, together with the one reported previously (light gray, pdb code: 5M4Y). (**C**) Stereo view of the 2F$_o$–F$_c$ electron density map contoured at 1.8σ level around the binding interface. Maps for Sec3 and Sso2 are colored in cyan and orange, respectively.

The online version of this article includes the following figure supplement(s) for figure 1:

**Figure supplement 1.** Constructs and purification of the Sso2/Sec3 complex.

transmembrane part, together with a shorter Sec3 PH domain (aa75–260) (*Figure 1A* and *Figure 1—figure supplement 1*). A stable binary complex was obtained via size exclusion chromatography (SEC) (*Figure 1—figure supplement 1*), which was then crystallized by hanging drop vapor diffusion. Diffraction data to 2.19 Å resolution was collected at the ESRF synchrotron site and the crystal structure was determined by the molecular replacement method. The crystal belongs to space group P1

**Table 1.** Data collection and refinement statistics.

| | |
|---|---|
| Wavelength (Å) | 0.9793 |
| Resolution range (Å) | 19.95–2.19 (2.27–2.19) |
| Space group | P1 |
| Unit cell (a, b, c; Å) ($\alpha$, $\beta$, $\gamma$; °) | 50.961, 58.402, 83.286 104.284, 98.494, 113.198 |
| Total reflections | 164,051 (15,510) |
| Unique reflections | 40,853 (3771) |
| Multiplicity | 4.0 (3.9) |
| Completeness (%) | 96.08 (89.79) |
| $I/\sigma(I)$ | 6.73 (1.18) |
| Wilson B-factor | 31.15 |
| R-*merge* | 0.1641 (1.094) |
| R-*meas* | 0.1894 (1.271) |
| R-*pim* | 0.09332 (0.636) |
| $CC_{1/2}$ | 0.99 (0.438) |
| CC* | 0.997 (0.781) |
| Reflections used in refinement | 40,587 (3771) |
| Reflections used for R-free | 2008 (181) |
| R-work | 0.1990 (0.2864) |
| R-free | 0.2394 (0.3374) |
| CC(work) | 0.943 (0.699) |
| CC(free) | 0.907 (0.601) |
| Number of non-hydrogen atoms | 5837 |
| Macromolecules | 5,419 |
| Solvent | 418 |
| Protein residues | 658 |
| RMS (bonds) | 0.002 |
| RMS (angles) | 0.50 |
| Ramachandran favored (%) | 97.82 |
| Ramachandran allowed (%) | 2.18 |
| Ramachandran outliers (%) | 0.00 |
| Rotamer outliers (%) | 0.00 |
| Clashscore | 5.76 |
| Average B-factor | 40.87 |
| Macromolecules | 40.74 |
| Solvent | 42.65 |

Statistics for the highest-resolution shell are shown in parentheses.

(a=50.96 Å, b=58.40 Å, c=83.29 Å; α=104.28°, β=98.49°, γ=113.20°). The final structure was refined to $R_{work}$ and $R_{free}$ of 19.9% and 23.9%, respectively, with an average B factor of 40.74 Å$^2$ for all macromolecules (*Table 1*).

The final model contains two copies of the Sso2/Sec3 complex per asymmetric unit, together with 418 ordered water molecules. One of the complexes contains residues 4–8, 33–149, and 197–226 of Sso2 and residues 76–250 of Sec3; the other contains residues 6–14, 33–148, and 196–226 of Sso2 and residues 76–250 of Sec3. Corresponding regions in the two complex structures are essentially identical, with r.m.s.d. of 0.45 Å for all aligned backbone atoms (*Figure 1B*). 2F$_o$–F$_c$ electron density maps have a high quality, and sidechains of most residues in both Sec3 and Sso2 can be confidently built (*Figure 1C*).

Primary sequence alignments of Sso2 homologs from various yeast species reveal two conserved three-residue motifs toward the N-terminus of Sso2, which we name as NPY motifs. These double NPY motifs are connected to the highly conserved core of Sso2 (i.e. Habc and H3) via a non-conserved linker with variable lengths in different homologs (*Figure 2A* and *Figure 2—figure supplement 1*). These NPY motifs bind individually to Sec3 in the two structural copies (*Figure 2B and E*). In one complex structure, residues 4–8 of Sso2 show clear densities in the 2F$_o$–F$_c$ map (*Figure 2C*) in the other structure we could unambiguously trace sidechains of residues 10–14, but for residues 6–9 we could build only the backbone atoms (*Figure 2F*). Despite variations in the flanking residues of these NPY motifs, the NPY cores adopt essentially the same conformation and share similar contacts with ordered solvent molecules and neighboring residues from Sec3 (*Figure 2—figure supplement 2*). Particularly, the tyrosine residues in both cases (i.e. Y7 and Y13) are docked into a conserved hydrophobic pocket on the concave surface on Sec3 (*Figure 2D and G*). Overall, the NPY motif adopts a T-shaped conformation, with its broad top part shaped by the asparagine (N) and the proline (P) residues, which is stabilized by a hydrogen bond between the carboxyl group of the asparagine and the amide proton of the proline (*Figure 2H*). The tyrosine residue sticks out to form a pin-like structure that fits neatly in the pocket on Sec3.

## The two NPY motifs of Sso2 binds synergistically to Sec3

To determine how the two NPY motifs of Sso2 interact with Sec3, we carried out isothermal titration calorimetry (ITC) assays using synthetic polypeptides of Sso2 and recombinant Sec3-PH purified from bacteria (*Figure 3*). Wild type (WT) double NPY motifs of Sso2 (aa1–15) bound Sec3 with a dissociation constant ($K_d$) of 21.1 µM (*Figure 3A*). However, each NPY motif individually bound Sec3 much more weakly, with binding affinities reduced by three- to fourfold (*Figure 3B and C*). Mutation of all core residues of the first NPY motif to alanines (M5) slightly reduced the binding affinity ($K_d$ = 34.9 µM) (*Figure 3D*), whereas mutation of the second NPY motif (M6) drastically affected the interaction, with a $K_d$ of 188 µM (*Figure 3E*). The synthetic polypeptide with both NPY motifs mutated (M7) displayed no detectable interaction with Sec3 (*Figure 3F*).

## Mutation of Sso2 NPY motifs inhibits cell growth as well as secretion of Bgl2 and invertase

To assess the in vivo role of the interactions between Sec3 and the two NPY motifs of Sso2, we constructed various *sso2* alanine substitution mutations in a yeast integrating vector and introduced them into a yeast strain deleted for the paralogous gene, *SSO1*. The mutations were incorporated into the endogenous *SSO2* locus by the loop-in/loop-out method, leaving the surrounding sequence entirely unaltered (*Figure 4A* and *Figure 4—figure supplement 1*; *Novick and Botstein, 1985*). The *sso1Δ sso2* mutants were tested for growth at both 25°C and 37°C (*Figure 4B* and *Figure 4—figure supplement 1*). No effect was observed with single, double, or triple mutations in the first NPY domain (*sso1Δ sso2M1-M4*), however changing all four residues to alanine (*sso1Δ sso2M5*) resulted in reduced growth at 37°C (*Figure 4—figure supplement 1*). Mutating all four residues of the second NPY domain to alanine (*sso1Δ sso2M6*) did not affect growth at any temperature, however changing all residues of both the first and second NPY domain to alanine (*sso1Δ sso2M7*) resulted in significantly reduced growth at both 25°C and 34°C and severely impaired growth at 37°C (*Figure 4B*). The synergistic effects of eliminating the first and second NPY motif of Sso2 suggest that both motifs are functional and at least partially redundant.

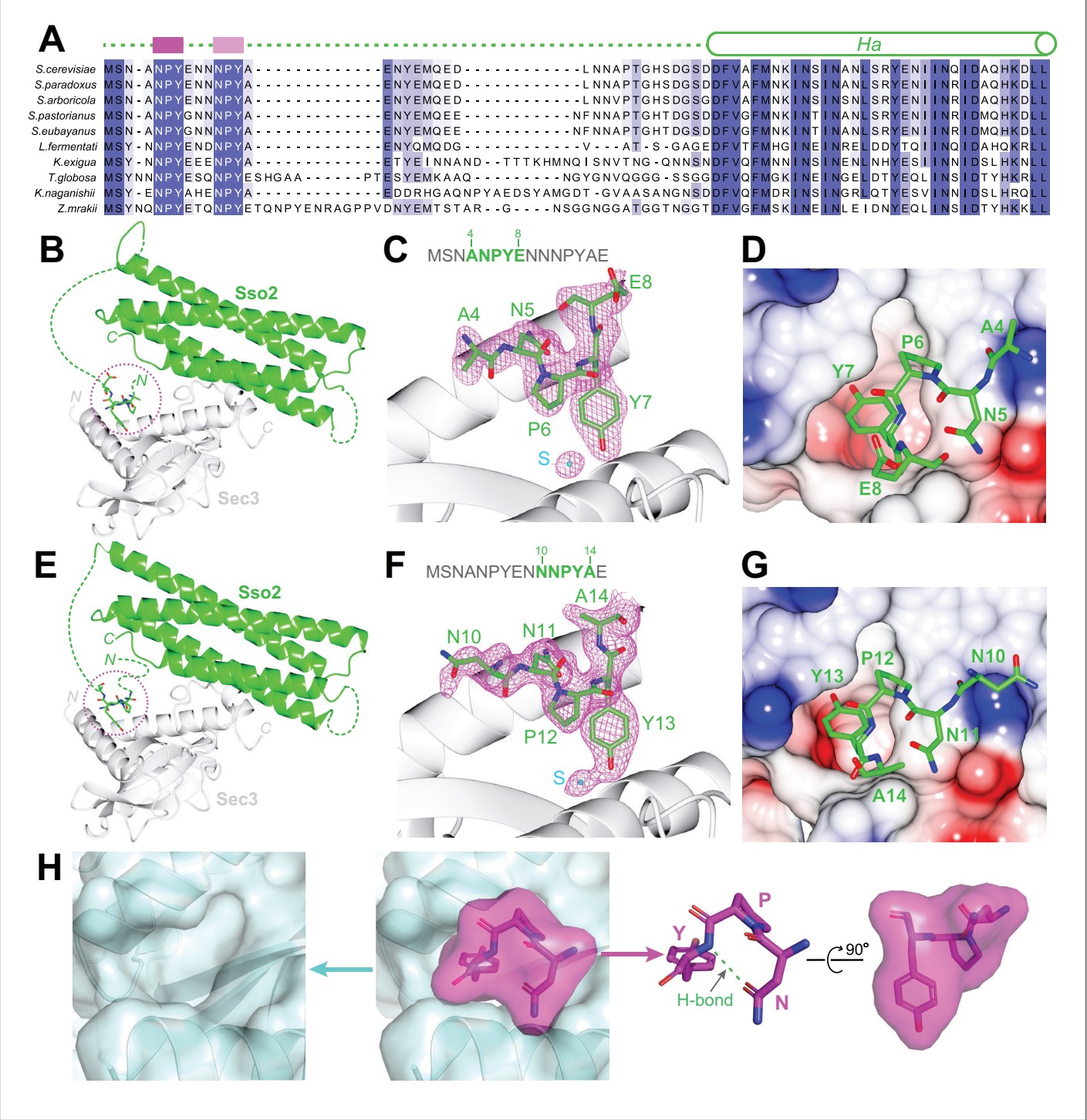

**Figure 2.** Structural analyses of the interaction between the NPY motifs of Sso2 and Sec3. (**A**) Sequence alignments of Sso2 homologs. Conserved residues are shaded in dark (highly conserved) or light blue (partially conserved). The two NPY motifs are marked as magenta blocks above the aligned sequences, which are connected to helix Ha via a variable linker. (**B**) Ribbon diagram of the crystal structure of the Sso2/Sec3 complex with first NPY motif (shown as sticks) of Sso2 bound to Sec3. (**C**) An enlarged view of the NPY motif in (**B**) together with the $2F_o$–$F_c$ map contoured at 1.5σ. (**D**) Sticks of the NPY motif on top of an electrostatic surface plot of Sec3. (**E–G**) Crystal structure shows how the second NPY motif of Sso2 interacts with Sec3. (**H**) Separate views of the binding site show the cork-like NPY motif (magenta) and the complementary cradle-like pocket on Sec3 (light blue). The structure is shown as semitransparent surface together with ribbon diagrams (Sec3) or sticks (Sso2). The broad top part of the 'cork' of Sso2 is stabilized by a hydrogen bond between the carboxyl group of the Asn (**N**) sidechain and the amide proton of Tyr (**Y**).

*Figure 2 continued on next page*

*Figure 2 continued*

The online version of this article includes the following figure supplement(s) for figure 2:

**Figure supplement 1.** Structure of the Sso2/Sec3 complex and conservation analysis.

**Figure supplement 2.** Local chemical environment around the NPY motifs in the Sso2/Sec3 complex structure.

We next assayed the export of two different cell surface enzymes, Bgl2 and invertase. Bgl2 is synthesized and secreted constitutively, while the synthesis of invertase is under hexose repression. Both enzymes become trapped at the cell surface by the cell wall glucan and this external pool can be released by treatment of cells with exogenous glucanase, while any internal pool remains associated with the resulting spheroplasts (*Yuan et al., 2017*). Using western blot analysis to measure the internal and external pools of Bgl2, we found that the secretory efficiency at 37°C generally paralleled growth: *sso1Δ sso2M1-M6* showed only a modest accumulation of an internal pool, while *sso1Δ sso2M7* accumulated a significantly larger internal pool (*Figure 4C and D* and *Figure 4—figure supplement 1*).

To assay invertase secretion, we started with cells grown at 25°C in 5% (w/v) glucose to repress synthesis and then shifted to 0.1% (w/v) glucose to derepress synthesis and simultaneously shifted the cells to 37°C. Using these conditions we found that *sso1Δ sso2M1-M4* and *sso1Δ sso2M6* were not significantly different from the *sso1Δ SSO2* control, while *sso1Δ sso2M5* showed a minor defect in invertase secretion and *sso1Δ sso2M7* showed a more substantial defect (*Figure 4E* and *Figure 4— figure supplement 1*).

## Mutations of the Sso2 NPY motifs cause polarized accumulation of secretory vesicles

Defects on the secretory pathway are typically associated with the accumulation of membrane bound intermediates (*Novick et al., 1980*). Loss of function of exocytic SNAREs, including Sso1 and Sso2, leads to the accumulation of secretory vesicles (*Aalto et al., 1993*). Secretory vesicles are normally delivered to sites of polarized cell surface growth, such as the tip of the bud, early in the cell cycle, and the neck separating the mother cell and bud, late in the cell cycle. Thin section electron microscopy revealed an accumulation of vesicles in the mutants that mirrored their growth: the *sso1Δ SSO2* control and the *sso1Δ sso2M6* mutant had relatively few vesicles per cell section, while *sso1Δ sso2M5* and *sso1Δ sso2M7* had many more (*Figure 5A–E*). The vesicles were similar in size in the different strains (*Figure 5F*) and were found preferentially within the buds of small budded cells (*Figure 5A–D*).

## Mutations in the NPY motifs of Sso2 affect Snc1 recycling

In addition to the export of newly synthesized cell surface proteins, secretory vesicles are also important for recycling certain plasma membrane proteins back to the cell surface after they have been internalized by endocytosis. The Snc1 v-SNARE has been shown to rapidly cycle from secretory vesicles to the plasma membrane, and then into endocytic vesicles from which it is recycled through the Golgi into a new round of secretory vesicles (*Lewis et al., 2000*). Under normal growth conditions, Snc1 is predominantly found on the plasma membrane, with only a minor pool in internal structures. Impeding any step in the cycle leads to a shift in the steady-state distribution of GFP-Snc1. We examined the distribution of GFP-Snc1 in various *sso1Δ sso2* mutants. In the *sso1Δ SSO2* control and the *sso1Δ sso2M6* mutant, GFP-Snc1 was mostly on the plasma membrane with a small number of internal, patch-like structures apparent (*Figure 6A and C*). In contrast, the *sso1Δ sso2M5* and *sso1Δ sso2M7* mutants showed a much greater fraction of cells with internal patches of GFP-Snc1, presumably representing concentrations of secretory vesicles, and a greatly reduced localization to the plasma membrane (*Figure 6B, D and E*).

## The *sso2* mutations have no effect on the actin-independent localization of Sec3

Secretory vesicles are delivered to sites of cell surface growth by the type V myosin, Myo2, moving along polarized actin cables (*Pruyne et al., 2004*). Loss of actin or Myo2 function leads to the rapid depolarization of a vesicle marker, such as the Rab GTPase Sec4 (*Salminen and Novick, 1989*). Sec3, in contrast, remains associated with the tips of small buds and the necks of large buds even after actin polymerization has been blocked by addition of Latrunculin A (LatA) (*Finger et al., 1998*). We

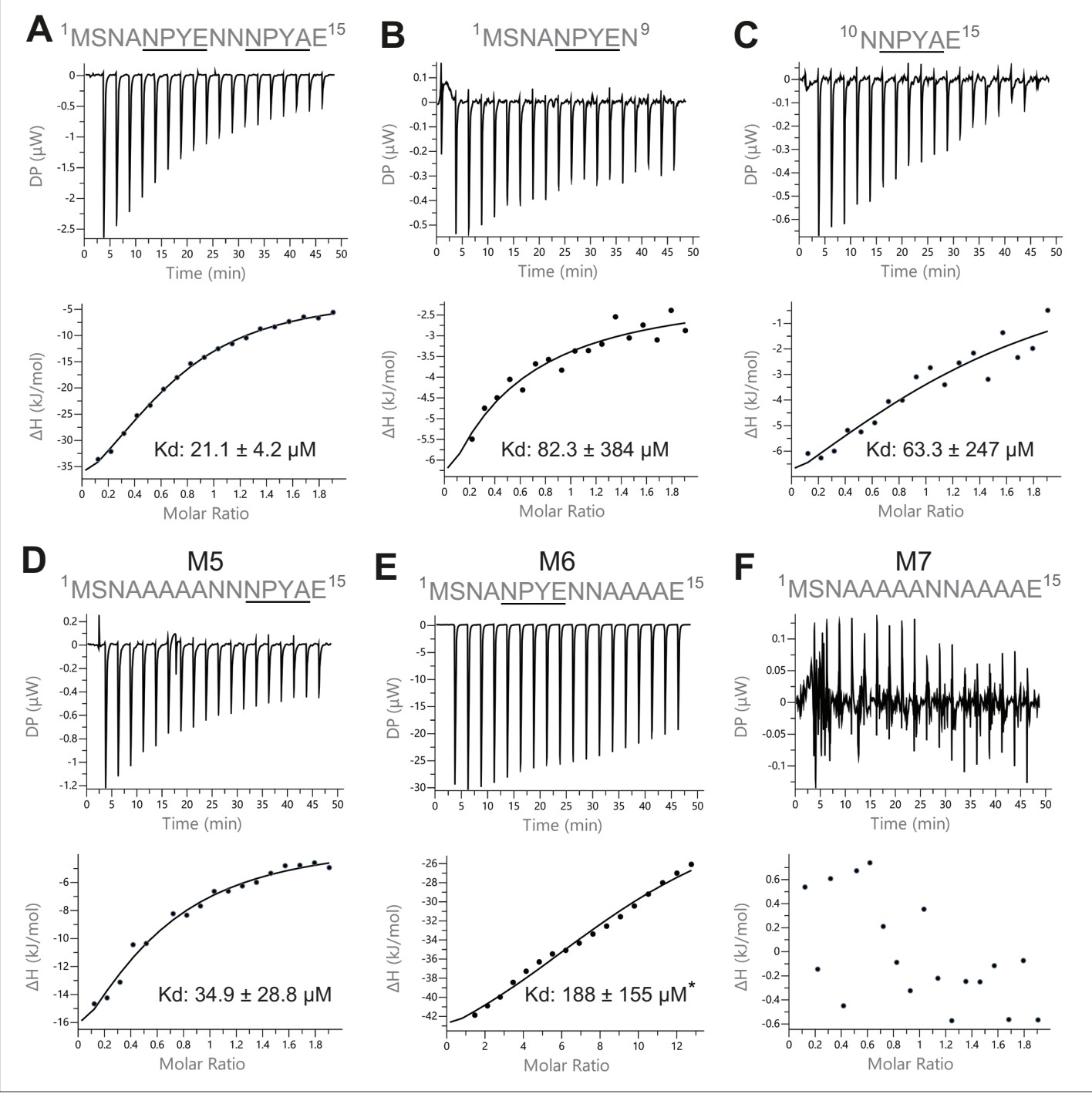

**Figure 3.** Isothermal titration calorimetry (ITC) measurements of the interaction between Sec3 and variant versions of the NPY motifs of Sso2. (**A**) Wild type double NPY motifs of Sso2 bound Sec3 with a dissociation constant ($K_d$) of approximately 21 µM. (**B** and **C**) Either of the two NPY motifs alone bound Sec3 much more weakly than the two together, with $K_d$ values increased by three- to fourfold. (**D** and **E**) Mutation of either NPY motif (i.e. **M5 and M6**) also substantially reduced the binding affinity of Sso2 to Sec3. (**F**) Double mutation (**M7**) of both NPY motifs completely abolished the interaction between Sso2 and Sec3. For M6 in (**E**), the peptide concentration had to be increased by sixfold to detect its very weak interaction with Sec3.

determined if the interactions between Sec3 and the NPY motifs of Sso2 are required for the actin-independent localization of Sec3. GFP-Sec4 and Sec3–3×GFP were expressed in *sso1Δ SSO2*, *sso1Δ sso2M5*, and *sso1Δ sso2M7* cells. Localization was evaluated after treatment with either LatA or DMSO for 15 min. The polarized localization of GFP-Sec4 was lost after treatment with LatA (*Figure 7A*

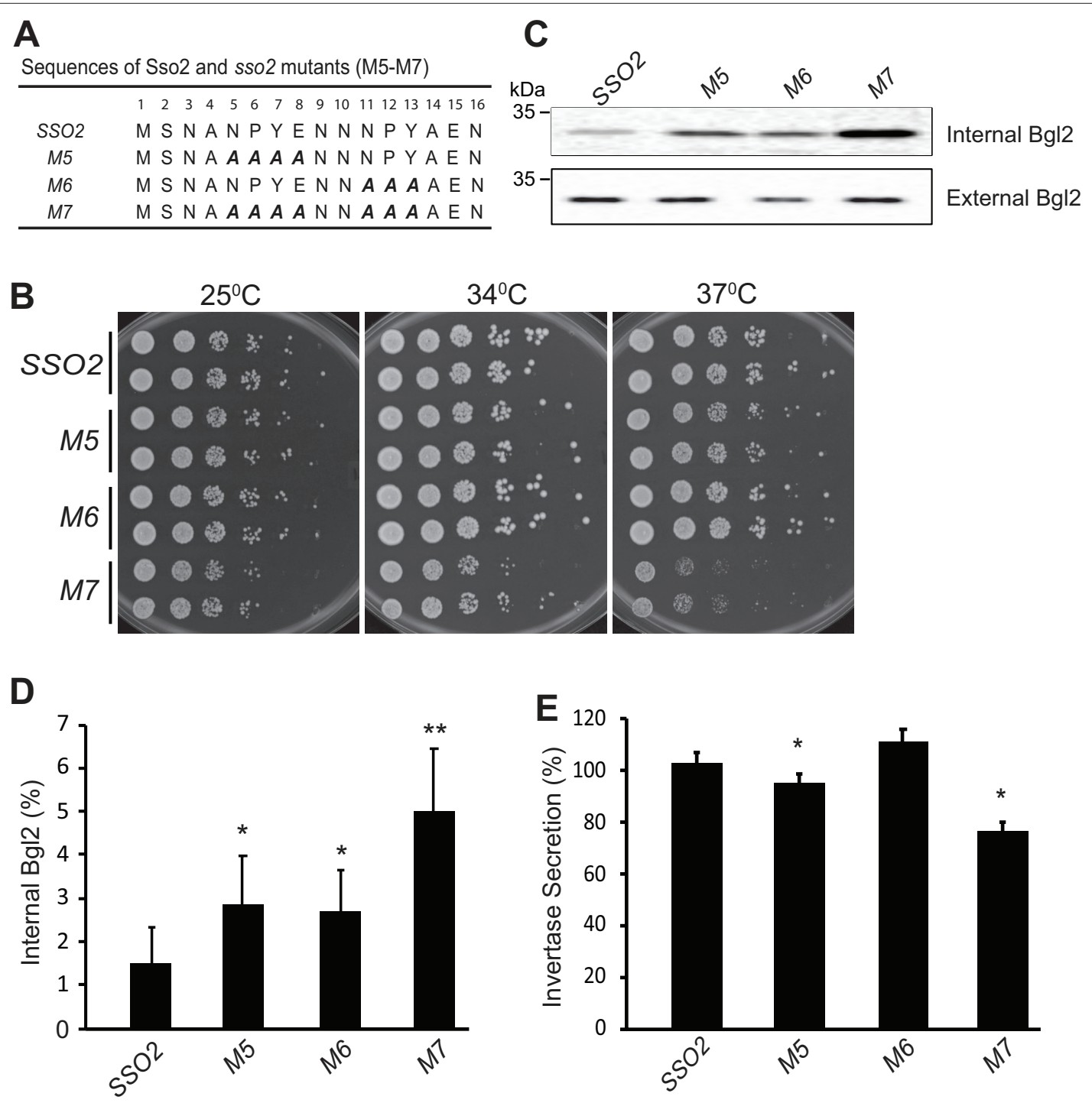

**Figure 4.** Mutations in *sso2* inhibit cell growth and secretion of Bgl2 and invertase. (**A**) Sequences of three *sso2* mutants generated by site-directed mutagenesis. Residues in the first and/or the second NPY motif that were mutated to Ala (**A**) are shown in bold italics. (**B**) *Sso2* mutants in an *sso1Δ* background partially inhibit cells growth at 37°C. Control *sso1⊗ SSO2* or *sso1Δ sso2* mutant cells were grown overnight in YPD medium. An aliquot (0.2 OD$_{600}$ units) of cells from each strain was collected, serially diluted by fivefold and spotted onto YPD plates. Plates were incubated at 25°C, 34°C, or 37°C for 2 days. (**C–D**) The *sso1Δ sso2* mutants were grown at 25°C in YPD medium overnight to early log phase and shifted to 37°C for 90 min. The internal and external fractions were prepared as described in *Materials and methods*. (**C**) The internal or external pools of Bgl2 were detected by western blotting. Several mutations in *SSO2* caused inhibition of Bgl2 secretion. (**D**) Quantitation of internal Bgl2 was determined by ImageJ. Results were analyzed based on seven independent experiments. Error bar represents SD, n=7. *p<0.005, **p<2e5. (**E**) Strains were grown at 25°C in YP containing

*Figure 4 continued on next page*

*Figure 4 continued*

5% (v/v) glucose medium overnight to early log phase. 1 $OD_{600}$ unit of cells was collected from each strain and shifted to YP containing 0.1% (v/v) glucose medium, and incubated at 37°C for 45 min. Four independent experiments were performed. Error bars represent SD, n=4. *p<0.005.

The online version of this article includes the following figure supplement(s) for figure 4:

**Figure supplement 1.** Growth effect and influence on Bgl2 and invertase secretion of sso2 mutants of its first NPY motif.

*and B*). While the localization of Sec3−3×GFP was largely resistant to LatA treatment in the control and *sso1Δ sso2M5* mutant cells, the polarization of Sec3−3×GFP in *sso1Δ sso2M7* was only slightly reduced by LatA treatment (*Figure 7C and D*). These results demonstrate that the NPY motifs of Sso2 do not play a major role in the recruitment of Sec3 to sites of polarized growth. Notably, prior studies have shown that the actin-independent localization of Sec3 involves its interaction with the Rho1 and Cdc42 GTPases (*Guo et al., 2001*; *Zhang et al., 2001*).

## The NPY motifs play an essential role in stabilizing the interaction between Sso2 and Sec3

In the structure with the second NPY motif bound to Sec3, we see extra electron densities beyond the bound NPY motif. However, the quality of the map in that part was poor and we could only build the main chains for residues 6–9 (*Figure 8A*). We found that this 'fuzzy' part upstream of the bound NPY motif is in close contact with the C-terminal tip of the SNARE motif (i.e. H3) of Sso2 (*Figure 8B*). To find out whether the NPY motifs are essential in stabilizing the interaction between Sso2 and Sec3, we carried out ITC experiments using either WT or M7 mutant of Sso2 with the Sec3 PH domain. Our results show that WT Sso2 bound Sec3 robustly, with $K_d$ ~ 2.7 µM (*Figure 8C*). However, the M7 mutant of Sso2 (aa1–270) did not interact with Sec3 at all (*Figure 8D*).

We further examined their interactions using two other independent methods. Our electrophoretic mobility shift assays (EMSA) show that more and more WT Sso2 shifted up to the complex band with increasing amounts of Sec3 in the mixtures. However, no complex was formed when we mixed the M7 mutant of Sso2 with Sec3 (*Figure 8—figure supplement 1*). Consistently, our further test using SEC also shows that Sec3 could form complex with only WT but not the M7 mutant of Sso2 (*Figure 8—figure supplement 1*).

To investigate whether the M7 mutant of Sso2 also affects its interaction with Sec3 in vivo, we carried out co-immunoprecipitation of Sec3−3×Flag and Sso2 using yeast cell lysate. Our results show that, in contrast to the clear signal of WT Sso2 pulled down by Sec3−3×Flag, the pull-down band for the M7 mutant was much weaker and near the level of the negative control in which the Sec3−3×Flag was absent (*Figure 8—figure supplement 2*).

## Discussion

Sec3 is a subunit of the octameric exocyst complex, a tethering factor that marks the docking site for secretory vesicles in exocytosis. Sec3 is recruited to these sites by binding to the membrane-anchored small GTPases Rho1/Cdc42 and phosphoinositides on the plasma membrane (*Guo et al., 2001*; *Zhang et al., 2001*; *Zhang et al., 2008*; *Yamashita et al., 2010*). Sso2, similar to its homologous t-SNARE in neuronal exocytosis, bears a C-terminal transmembrane helix. We have previously reported crystal structures of the Sec3 N-terminal PH domain in complex with the closed form of Sso2 as a four-helix bundle (*Yue et al., 2017*). We found that Sec3 destabilizes the linker between Hc and H3 of Sso2 via an allosteric effect, which promotes the assembly of the binary complex between Sso2 and the other t-SNARE, Sec9, and thus drastically accelerates subsequent full SNARE assembly with the v-SNARE Snc1/2 to drive fusion of secretory vesicles with the plasma membrane. However, it remains unknown whether the conserved N-terminal extension of Sso2 also participates in the Sso2/Sec3 interaction and how Sso2 is effectively recruited to Sec3.

Here, we report a new crystal structure of the Sec3 PH domain in complex with a nearly full-length version of Sso2 that contains all sequence except for the C-terminal transmembrane helix. In addition to the interaction between the helical bundle of Sso2 and Sec3 observed in the previous study, we found an extra interaction interface between two conserved NPY motifs at the N-terminal end of Sso2 and a hydrophobic pocket on Sec3 (*Figure 1*). The NPY motifs are connected to the highly conserved

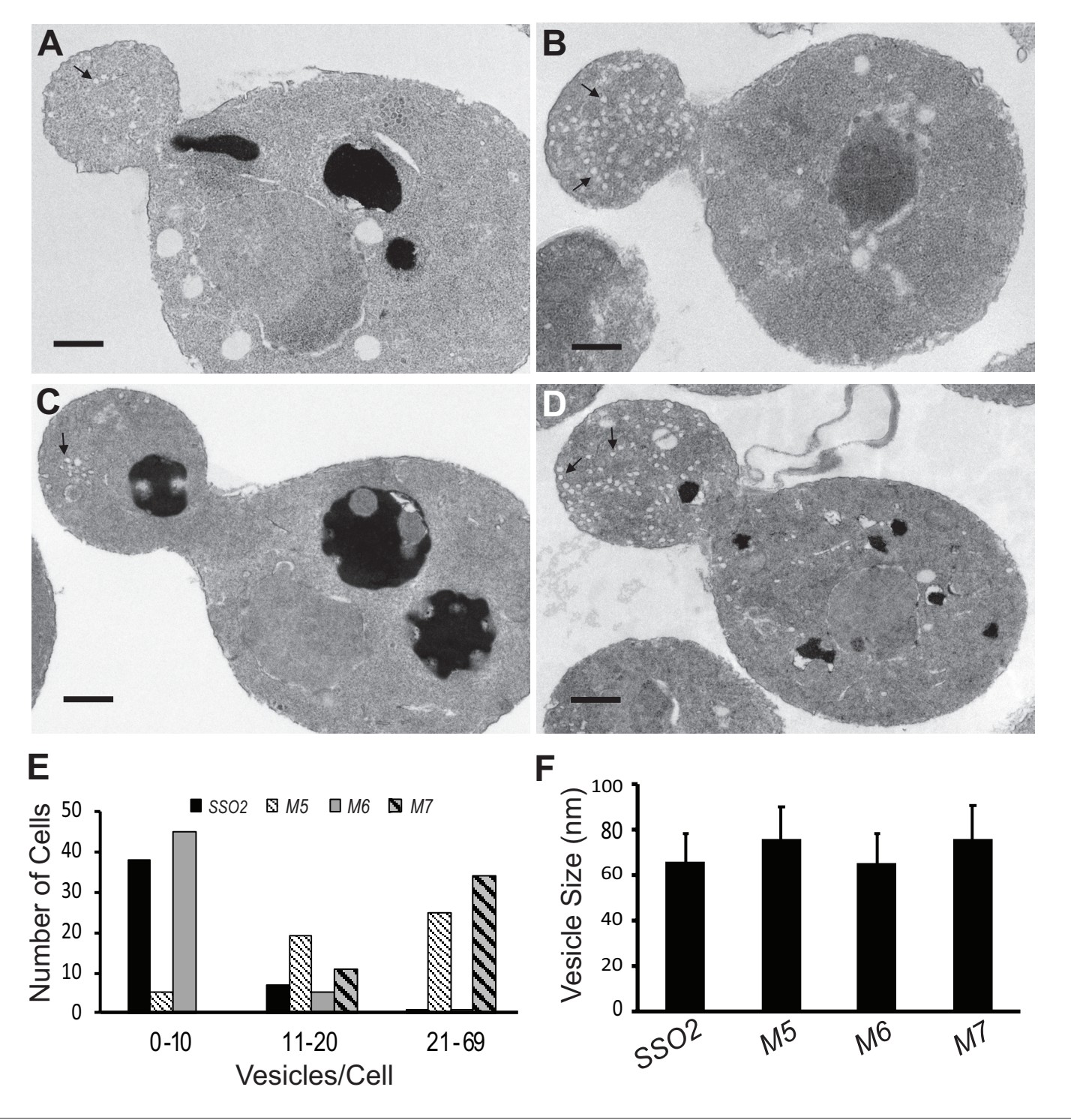

**Figure 5.** Thin section electron microscopy (EM) analysis shows polarized accumulation of secretory vesicles in the *sso2M5 and sso2M7* strains. A control *sso1Δ SSO2* strain and *sso1Δ sso2* mutants were grown in YPD medium at 25°C to an $OD_{600}$ of 0.5, then 10 $OD_{600}$ units of cells were collected and processed for EM analysis: (**A**) *sso1Δ SSO2*, (**B**) *sso1Δ sso2M5*, (**C**) *sso1Δ sso2M6*, and (**D**) *sso1Δ sso2M7*. Scale bar, 0.5 μm. *sso2M5* and *sso2M7* cells contain more secretory vesicles in the bud compared to *SSO2* and *sso2M6* cells. (**E**) Quantitation of secretory vesicles. The number of vesicles/cell was scored in 46 control cells, 49 *sso2M5 cells*, 51 *sso2M6* cells, and 45 *sso2M7* cells. (**F**) Measurement of vesicle size. The measurement was analyzed using ImageJ; 23–91 vesicles were measured.

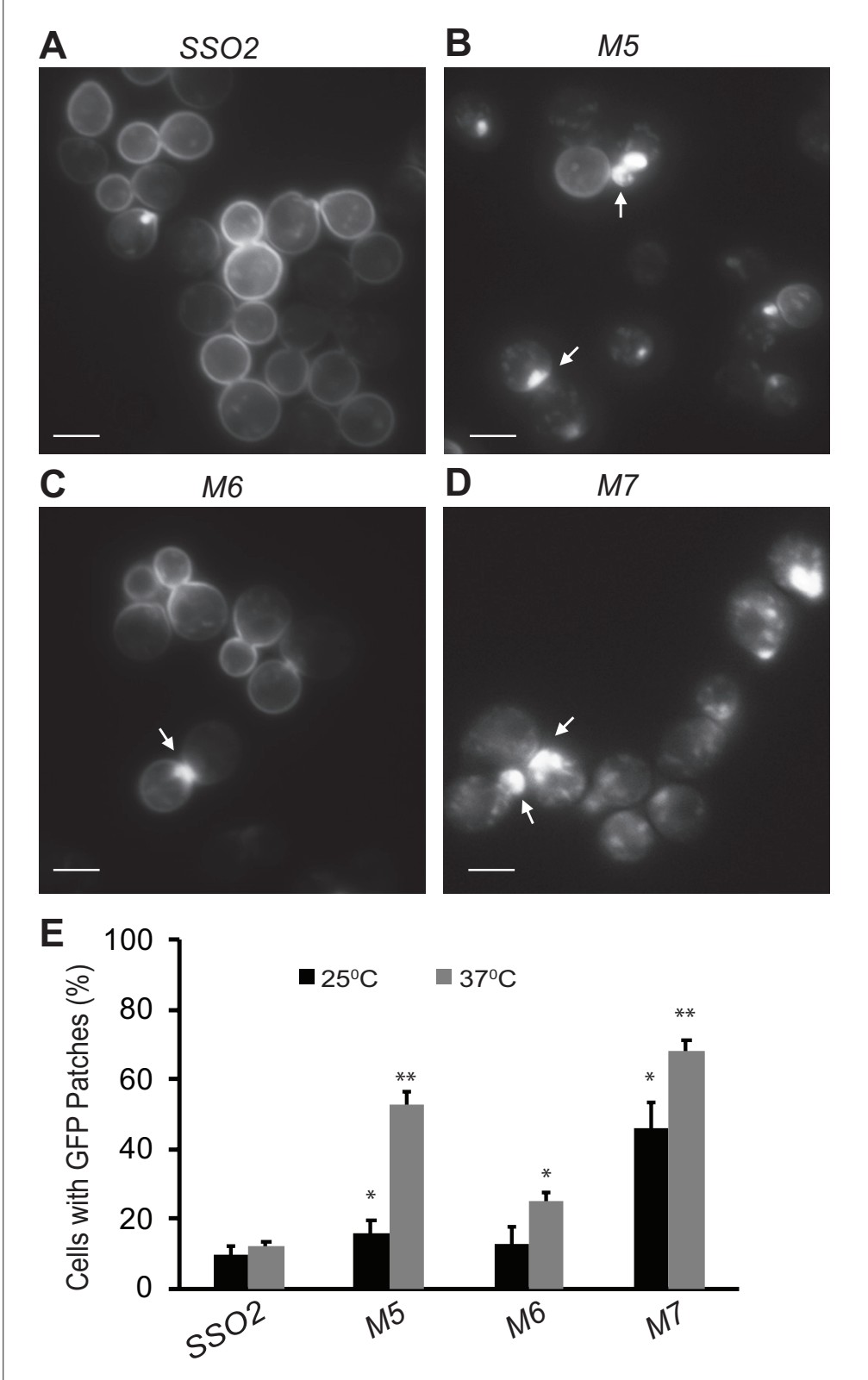

**Figure 6.** Snc1 recycling is affected in *sso2* mutants. (**A–D**) Representative images of *sso1Δ* SSO2 control, *sso1Δ sso2M5 sso1Δ sso2M6,* and *sso1Δ sso2M7* mutant cells. Strains harboring a CEN plasmid expressing GFP-Snc1 were grown in SC-Ura medium overnight at 25°C reaching an $OD_{600}$ around 0.5. Cells were further diluted and shifted to 37°C for 90 min prior to imaging. GFP-Snc1 localized predominantly to the PM in *SSO2* control cells and

*Figure 6 continued on next page*

*Figure 6 continued*

*sso2M6* cells yet formed GFP patches (indicated by arrows) near small buds or bud necks in *sso2M5* and *sso2M7* mutant cells. Scale bar, 5 µm. (**E**) Quantitation of cells with GFP-Snc1 patches. GFP-Snc1 patches near bud tips or necks were scored. Error bar represents SD, n=3. *p<0.05, **p<0.005.

helical core of Sso2 via a non-conserved linker with variable lengths (roughly 15–40 residues), which is invisible in our crystal structure, suggesting that it is mobile within the crystal structure.

There are two copies of the Sso2/Sec3 complex in our crystal structure. The two NPY motifs were independently bound to Sec3 in these two complex structures (*Figure 2*). Despite variations in the flanking residues, the cores of the two motifs, that is, residues Asn, Tyr, and Pro, adopt essentially the same conformation and have similar hydrogen bond networks with the neighboring Sec3 residues and water molecules (*Figure 2—figure supplement 2*). Our ITC experiments show that either of the two NPY motifs alone could bind Sec3 tightly, although with binding affinities three- to fourfold lower than the polypeptide with both motifs (*Figure 3*). Similarly, mutation of either motif to alanine also substantially reduced the interaction of the N-terminal part of Sso2 with Sec3, whereas simultaneous

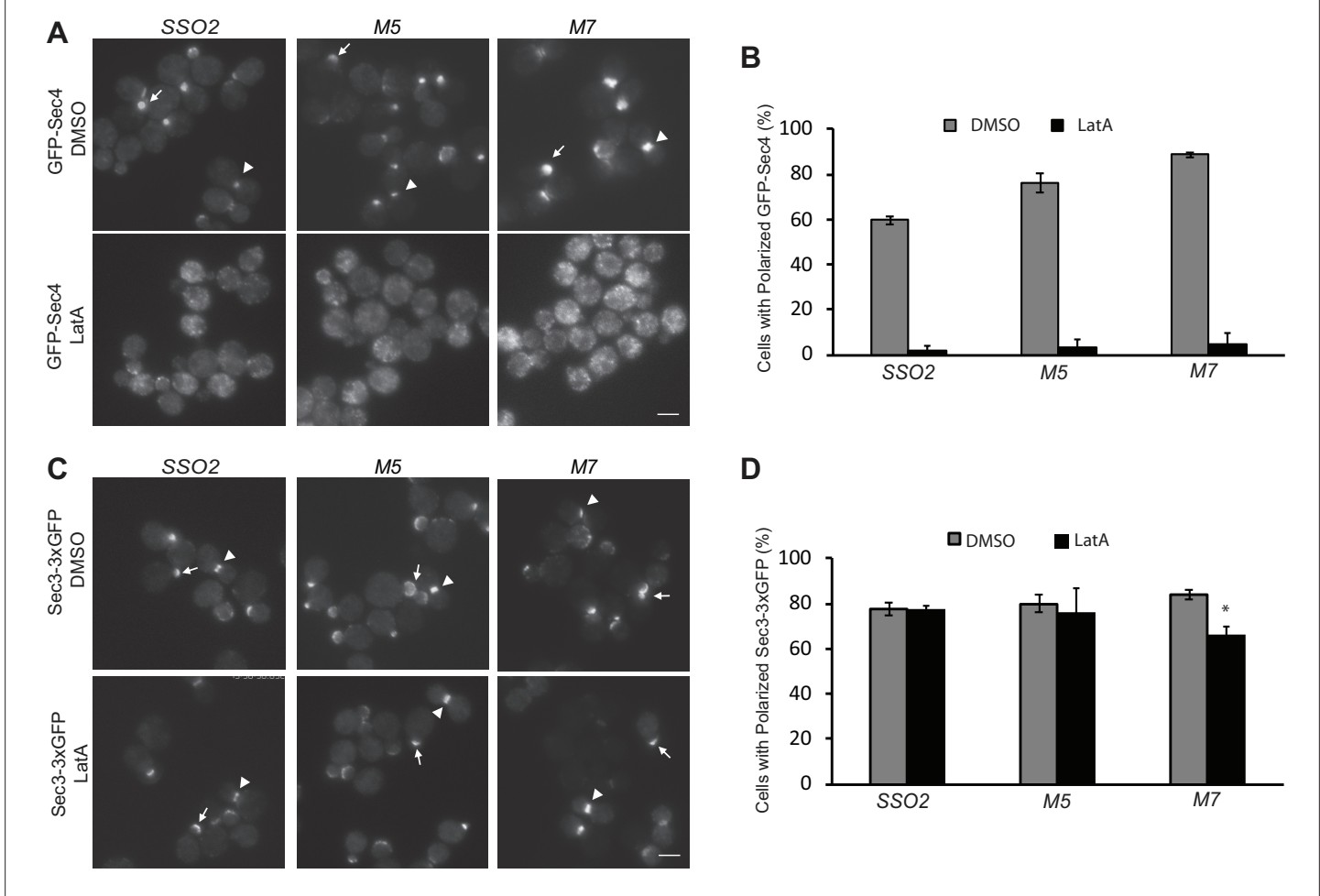

**Figure 7.** The *sso2* mutations have little effect on the actin-independent localization of Sec3. (**A**) A control *sso1Δ SSO2* strain or *sso1Δ sso2* mutants expressing Sec4-GFP were grown to early log phase in YPD at 25°C, then 1 OD$_{600}$ unit cells was collected and resuspended in 50 µl SC medium and incubated with 200 µM Latrunculin A (LatA) or DMSO at 25°C for 15 min prior to imaging. Images were captured by fluorescence microscopy. Sec4 is normally localized near the tip of small buds (arrow) and the neck of large buds (arrowhead) in DMSO-treated cells. Sec4 localization was disrupted in LatA-treated cells. Scale bar, 5 µm (for all images). (**B**) Quantitation of polarized Sec4-GFP. (**C**) A control *sso1Δ SSO2* strain and the *sso1Δ sso2* mutant strains expressing Sec3–3×GFP were grown and treated with LatA or DMSO under the conditions as described in A. Sec3 localization remains polarized after LatA treatment, either at the tips of small bus (arrow) or the necks of large buds (arrow head). Scale bar, 5 µm. (**D**) Quantitation of polarized Sec3–3×GFP. Error bars represent SDs based on three independent experiments in each case.

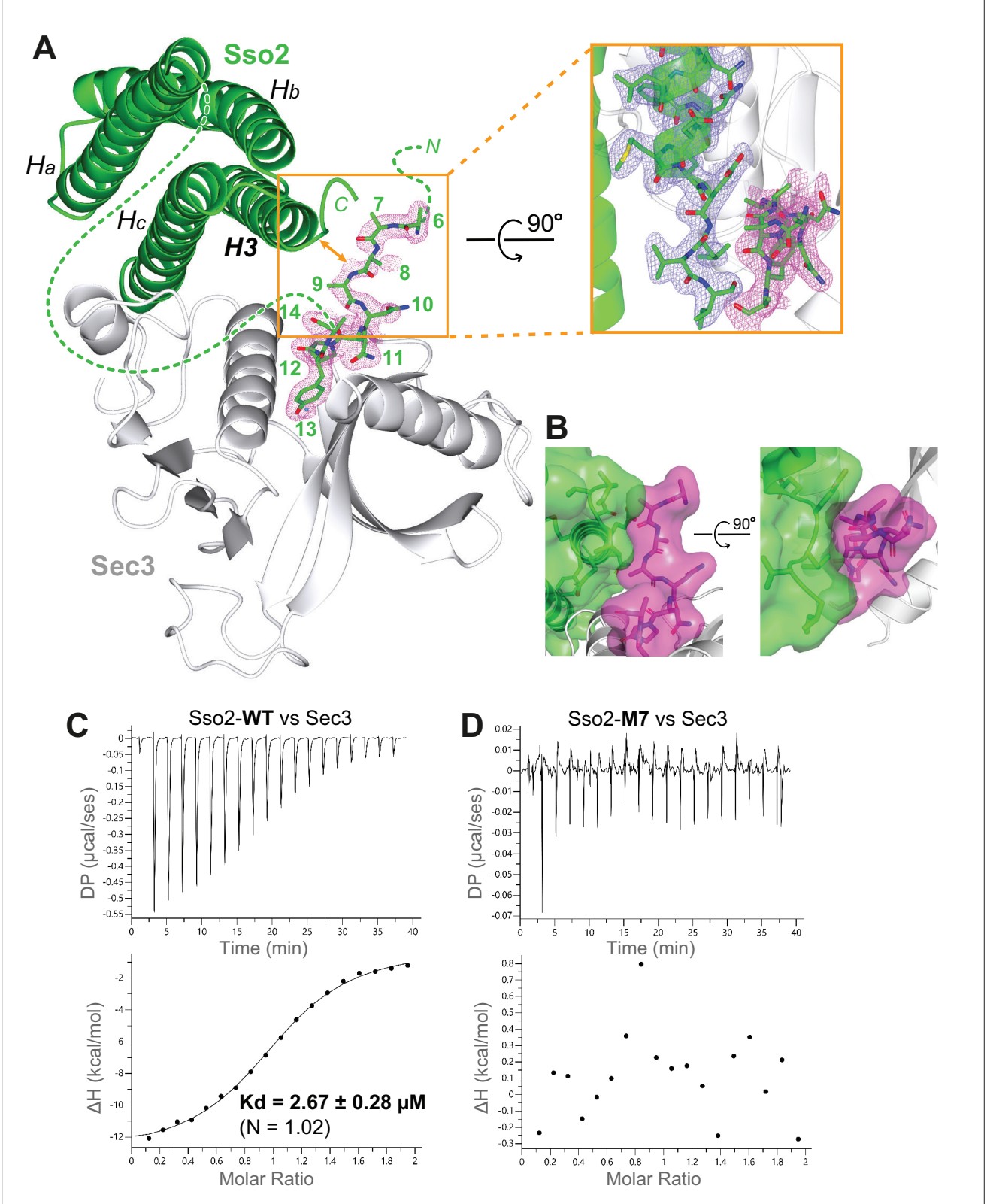

**Figure 8.** The NPY motifs of Sso2 are packed against the C-terminal tip of its soluble *N*-ethylmaleimide-sensitive factor-attachment protein receptor (SNARE) motif in the complex structure. (**A**) Ribbon diagram of the Sso2/Sec3 complex structure with the $2F_o-F_c$ map around the NPY motif (residue positions are marked by numbers) shown in dots (1.5σ). The enlarged view shows the close contact between the NPY motif and the C-terminal tip of the H3 helix of Sso2 with their $2F_o-F_c$ maps shown in magenta and blue, respectively. (**B**) Two orthogonal views of the contact site between the N-terminal

*Figure 8 continued on next page*

*Figure 8 continued*

extension (magenta) and the C-terminal region (green) of Sso2. Sso2 structure is shown as ribbons and sticks together with a semitransparent surface plot. (**C**) Isothermal titration calorimetry (ITC) result shows that wild type (WT) Sso2 (aa1–270) bound Sec3-PH robustly, with a $K_d$ of 2.67 µM. (**D**) Mutant M7 of Sso2 (aa1–270) did not bind the Sec3 PH domain.

The online version of this article includes the following figure supplement(s) for figure 8:

**Figure supplement 1.** In vitro binding assays to check the interaction between Sec3 and WT or mutant M7 of Sso2.

**Figure supplement 2.** Co-immunoprecipitation of Sec3-3×Flag and Sso2 from yeast extract.

mutation of both motifs completely abolished the interaction. Together, these data suggest that the two NPY motifs bind Sec3 synergistically.

To further understand the function of the NPY motifs in exocytosis, we carried out a series of in vivo assays using yeast strains carrying mutations of Sso2 in its NPY motifs. While mutation of either NPY motif alone only slightly reduced cell growth, simultaneous mutation of both NPY motifs severely impaired cell growth, particularly at 37°C (*Figure 4B*). We also explored how mutations of the NPY motifs of Sso2 influence protein secretion using Bgl2 and invertase as reporters. Our data show that mutation of both NPY motifs simultaneously substantially affects cell secretion efficiency, with a significantly larger internal pool of Bgl2 failing to reach the cell surface, whereas mutation of either motif alone yielded only a modest accumulation of an internal pool (*Figure 4C and D* and *Figure 4—figure supplement 1*). Similar effects were observed in invertase secretion, where the double NPY mutant M7 showed a more substantial defect than all other mutants (*Figure 4E* and *Figure 4—figure supplement 1*).

We further checked how the *sso2* NPY mutations affect fusion of secretory vesicles to the target sites on the plasma membrane. Mutation of both NPY motifs resulted in the accumulation of many vesicles within the cell. Similar results were also observed for the mutation of the first NPY motif, whereas that of the second NPY motif showed no significant effect (*Figure 5*). Consistently, we found that both the double NPY mutation and mutation of the first NPY motif caused accumulation of GFP-Snc1 patches within the cytoplasm (*Figure 6*). Taken together, we conclude that the NPY motifs of Sso2 play an essential role in the secretory pathway. Notably, however, the interaction between Sec3 and the NPY motifs of Sso2 is dispensable for the actin-independent localization of Sec3 (*Figure 7*), which is consistent with previous reports that recruitment of Sec3 to the plasma membrane is determined by its interaction with the small GTPases Rho1 and Cdc42 as well as phosphoinositides on the membrane (*Guo et al., 2001*; *Zhang et al., 2001*; *Zhang et al., 2008*; *Yamashita et al., 2010*).

Although the M7 mutant in the synthetic N-terminal part of Sso2 (aa1–15) disrupts its interaction with Sec3, the major interaction interface between the helical bundle of Sso2 and the Sec3 PH domain remains unchanged (*Figure 1B*). Therefore, it was intriguing to us why the M7 mutant showed such a strong deleterious effect in vesicle trafficking. Our ITC data reveal that the M7 mutant of Sso2 (aa1–270) completely abolished its interaction with the Sec3 PH domain (*Figure 8C and D*). The disrupted binding of the M7 mutant with Sec3 was further confirmed by two other in vitro experiments using purified recombinant proteins (*Figure 8—figure supplement 1*), as well as by our co-immunoprecipitation data (*Figure 8—figure supplement 2*). All these demonstrate that the NPY motifs play an essential role in stabilizing the interaction between Sso2 and Sec3. This might be explained by what was observed in the crystal structure of the Sso2/Sec3 complex, where the NPY motifs are in close contact with the C-terminal tip of the SNARE motif (i.e. H3) of Sso2 and may thus stabilize their interaction as explained below (*Figure 8A and B*).

The dual interaction interfaces between Sso2 and Sec3 are reminiscent of what has been seen in the structures of syntaxin-1 and Tlg2 in complex with their partner SM proteins Munc18 and Vps45, respectively (*Figure 9A–C*). In the latter two complex structures, a short peptide at the N-terminus of Munc-18 and Vps45, which was named 'N-peptide', forms a short helix and binds distally to the backside of the first domain of Munc18 or Vps45, opposite to their interaction sites with the helical bundles of the SNARE proteins (*Eisemann et al., 2020*; *Burkhardt et al., 2008*). In contrast, the binding site for the NPY motifs of Sso2 is on the same side of the Sec3 PH domain as the aforementioned direct contact with the C-terminal part of H3 of Sso2 (*Figure 9D*). Furthermore, the Sec3 PH domain is a single globular domain and much smaller than Munc18 and Vps45, both of which contain three domains that are folded into a horseshoe-like conformation. The helical bundle of syntaxin-1

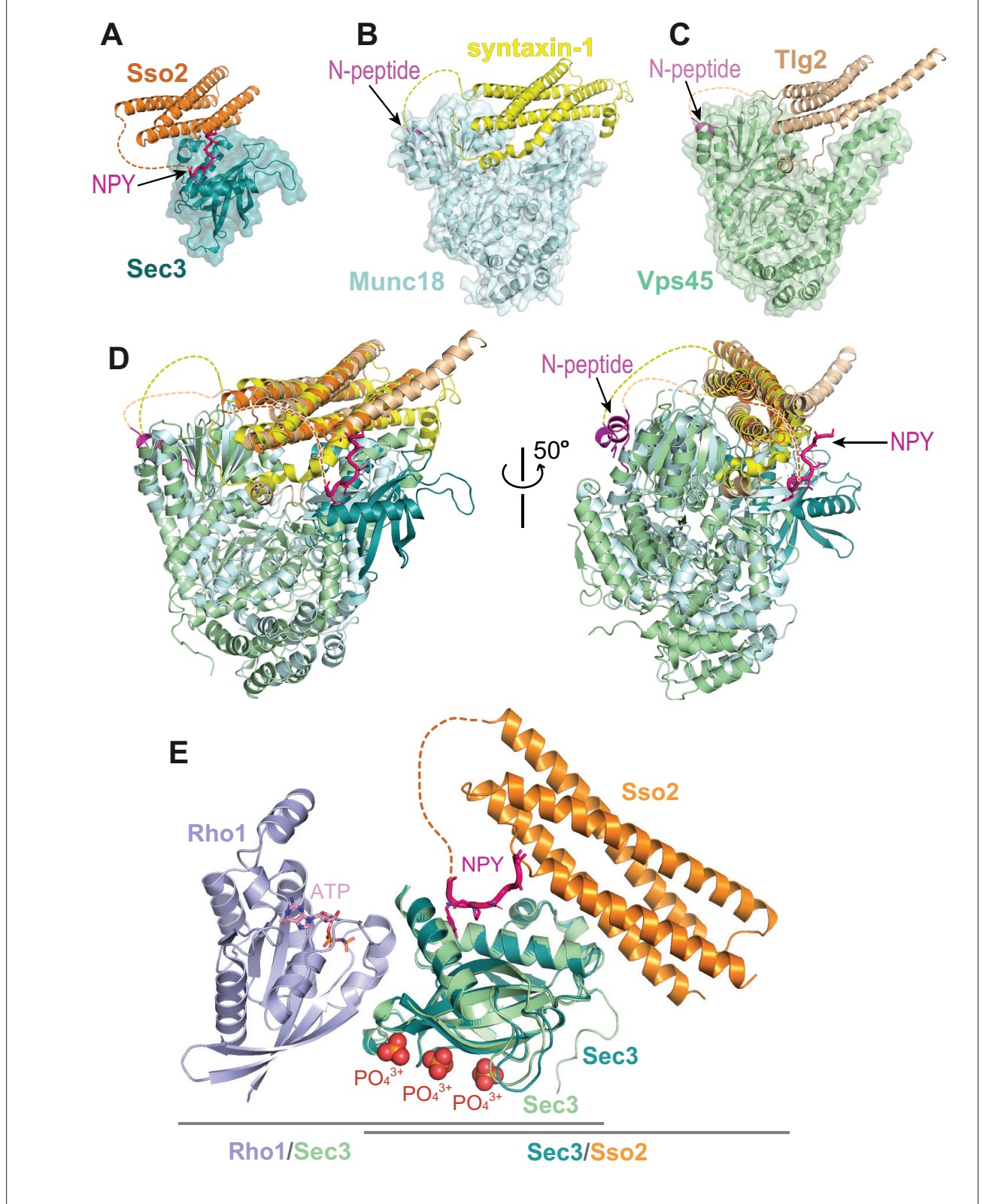

**Figure 9.** Interaction of the N-terminus of Sso2 with Sec3 in comparison with that in other syntaxin/SM protein complexes. (**A–C**) Crystal structures of the complexes of Sso2/Sec3 (PDB code: 7Q83), syntaxin-1/Munc18 (PDB code: 3C98), and Tlg2/Vps45 (PDB code: 2XHE), respectively. All N-terminal extensions of these t-SANREs are indicated by arrows. (**D**) Superposition of the three complex structures shown in A, B, and C. (**E**) Superposition of the Rho1/Sec3 complex (PDB code: 3A58) with the Sec3/Sso2 complex (PDB code: 7Q83) on top of their overlapped Sec3 components.

*Figure 9 continued on next page*

*Figure 9 continued*

The online version of this article includes the following figure supplement(s) for figure 9:

**Figure supplement 1.** A hypothetical working model for how Sso2 recruits Sec3 in vesicle docking.

and Tlg2 is clamped between the two tips of the 'horseshoe'. Additionally, in those two complex structures, the C-terminal extensions of the SNARE motifs (i.e. H3) form a short helix and insert into a hydrophobic pocket deeply inside the 'horseshoe' (*Figure 9B and C*). These together ensure a strong interaction between the two t-SNAREs and their partner SM proteins. In contrast, the interaction interface between Sec3 and Sso2 is very small, and thus their interaction is presumably much weaker than that in the Munc18/syntaxin-1 and Vps45/Tlg2 complexes. Our ITC data show that double mutation of both NPY motifs completely abolished the interaction between Sso2 and Sec3 (*Figure 8C and D*). Consistently, our in vivo data demonstrate that simultaneous mutation of both NPY motifs substantially impaired protein secretion as well as recycling of cell surface proteins. We therefore think that the NPY motifs play an important role in stabilizing the interaction of Sso2 with Sec3 by providing an additional binding site, which ensures effective recruitment of Sso2 to vesicle-docking sites on the plasma membrane.

Given that binding of Sec3 destabilizes the linker connecting H3 to Hc and prepares H3 for its subsequent interaction with Sec9, we think that another possible role for the NPY motifs, which are packed tightly against the C-terminal tip of the SNARE motif H3 (*Figure 8A and B*), might be to serve as a stopper to hold the destabilized SNARE motif in place to prevent a premature release of H3 before the right time comes for it to form a binary complex with the other t-SNARE protein Sec9.

Notably, the N-peptide motif is ubiquitously present in all syntaxins that interact with SM proteins (*Latham et al., 2006*; *McEwen and Kaplan, 2008*; *Johnson et al., 2009*; *Khvotchev et al., 2007*). It serves as an initiator to recruit SM proteins to their SNARE partners to facilitate their subsequent assembly (*Rathore et al., 2010*). Given that the NPY motifs are connected to the rest of Sso2 via a long variable linker, we hypothesize that they may similarly function like fishing hooks to search for Sec3 around the membrane-anchored Sso2 (*Figure 9—figure supplement 1*). Once the 'hooks' find and bind to Sec3, Sso2 would be locally restrained, which would promote the binding of Sec3 to the helical bundle of Sso2. This would in turn lead to the destabilization of the linker between Hc and H3 of Sso2 and thus promote the assembly of the binary complex between Sso2 and Sec9 to facilitate the final formation of the full SNARE complex with the v-SNARE Snc1/2, which eventually drives the fusion of secretory vesicles with the plasma membrane.

## Materials and methods

### Molecular cloning of expression constructs for in vitro assays

The Sso2 sequence excluding only its C-terminal transmembrane domain (aa1–270) was sub-cloned into the pET-15b vector (Novagen) between the *Nde*I and *Bam*HI sites using the following two primers: 5'-GGCAGCCATATGATGAGCAAC-3' (forward) and 5'-GACCGGATCCTTAACATCTTATTT TG-3' (reverse). The Sec3 PH domain sequence (aa75–260) was sub-cloned into pET-15b between the *Bam*HI and *Xho*I sites using forward primer 5'-GCTGGGATCCTCGAATTTTTTAGCCGAACAATATG-3' and reverse primer 5'-GCTGCTCGAGTTATGTGATGACTGCTCTTTGATAAC-3'.The pET15b plasmid provides an N-terminal His$_6$ tag followed by a thrombin cleavage site prior to the target proteins. All constructs were validated by DNA sequencing.

### Protein expression and purification

*Escherichia coli* strain BL21(DE3) cells harboring the expression plasmids for Sso2 and Sec3 were cultured in Luria Broth (LB) medium containing 50 mM ampicillin at 37°C to an OD$_{600}$ of 0.6–0.8. Over-expression of the target proteins was induced using 0.5 mM isopropylthio-β-D-galactoside and cultures were incubated at 18°C overnight. Cells were harvested by centrifugation (6000× *g*, 12 min, 4°C). The pellets were resuspended in pre-chilled lysis buffer containing 20 mM HEPES (pH 7.5), 100 mM NaCl, 20 mM imidazole, and 10 mM β-mercaptoethanol. Cell were lysed using an Emulsi-Flex-C3 homogenizer (Avestin). After centrifugation (25,000× *g*, 40 min, 4°C) to remove cell debris, the supernatant was filtered through a 0.45 µm pore size filter and then loaded onto a 5 ml Ni-HiTrap

column (GE Healthcare) that had been pre-equilibrated in the same lysis buffer. After washing with 5 column volumes (cv) of lysis buffer, bound protein was eluted using a linear gradient concentration of imidazole in the lysis buffer (20–600 mM, ×25 cv). Elution fractions were checked on SDS-PAGE gels and those containing target proteins were pooled. The N-terminal His$_6$ tag was removed by incubating the purified proteins with ~3% (w/w) thrombin (4°C, overnight) and then subjected to SEC using a Superdex S-200 16/60 column (GE Healthcare) pre-equilibrated with the running buffer containing 20 mM HEPES (pH 7.5), 100 mM NaCl, and 1 mM DTT. Fractions of interest were pooled for later use.

To generate protein complex for crystallization, purified domains of Sec3 and Sso2 were mixed in a molar ratio of 2:1. After incubation at 4°C for 1 hr, the mixture was loaded on a Superdex S-200 16/60 column (GE Healthcare) pre-equilibrated with the same running buffer as above. Elution fractions were checked on a 15% (w/v) SDS-PAGE gel. The first elution peak containing both proteins were pooled and concentrated to 10–15 mg/ml using Amicon Ultra Centrifugal Filter Units (Millipore) with 10 kDa molecular weight cutoffs.

## Crystallization, data collection, and structure determination

Concentrated protein of the Sec3-Sso2 complex (~12 mg/ml) was subjected to extensive crystallization screening trials using commercial crystallization kits. Initial crystallization screenings were carried out at 22°C by the sitting drop vapor diffusion method using the Phoenix HT liquid handling robot (Rigaku) to set up dual droplets for each condition with drop volume of 0.2 and 0.3 µl (1:1 and 2:1, protein vs. reservoir solution) on 96-well sitting drop crystallization plates (Molecular Dimensions). The crystal used for final data collection was grown from a manually setup droplet containing 2 µl of protein and 1 µl of reservoir solution. Needle-shaped crystals of ~10 × 10 × 100 µm³ were obtained in a condition containing 0.1 M sodium acetate (pH 5.0), 0.2 M ammonium acetate, and 30% (w/v) PEG 4000. All crystals were harvested with nylon loops (Hampton Research) by flash-cooling in liquid nitrogen using the same reservoir solutions containing 20% (v/v) glycerol as cryo-protectant. X-ray diffraction data were collected at the beamline ID23-1 of the European Synchrotron Radiation Facility (ESRF) in Grenoble, France. Data reduction was carried out using the XDS program (*Kabsch and Crystallogr, 2010*).

For structure determination, the maximum-likelihood molecular replacement by PHASER (*McCoy et al., 2007*) was conducted using our previously determined structures of Sec3 and Sso2 (PDB code: 5M4Y) as the searching model (*Yue et al., 2017*). The structural models were carefully checked; all different regions as well as extra parts in the models where electron densities were clearly visible were manually built using the program COOT (*Emsley and Cowtan, 2004*). Refinement was carried out by Phenix.refine (*Adams et al., 2010*) using data of 20–2.19 Å. All subsequent structure analyses and figure generations were carried out using Pymol (http://www.pymol.org). The details of data collection and refinement statistics are summarized in *Supplementary file 1*.

## Isothermal titration calorimetry

All ITC measurements were conducted on a MicroCal PEAQ-ITC microcalorimeter (Malvern Panalytical). Purified Sec3 (aa75–260) and WT or M7 mutant Sso2 (aa1–270) proteins were dialyzed overnight against a buffer composed of 20 mM HEPES (pH 7.5), 100 mM NaCl, and 1 mM DTT. Synthetic Sso2 polypeptides were dissolved in the same buffer and their concentrations were determined using the DS-11+ Spectrophotometer (DeNovix). For ITC measurements of Sec3 with various Sso2 poplypeptides, the reaction chamber and the injection syringe contained 250 µl of 59 µM Sec3 and 45 µl of 590 µM poplypeptides, respectively. For ITC measurements of Sec3-PH with Sso2 (aa1–270), the reaction chamber held 300 µl of purified Sec3 (30 µM), while the injection syringe contained 75 µl of WT or MT Sso2 (75 µM). All titration experiments consisted of one initial 0.4 µl injection followed by 18 consecutive 2 µl injections with a duration of 4 s each and an interval of 120 s between two consecutive injections. The resulting data were analyzed with the MicroCal PEAQ-ITC Analysis Software (Malvern Panalytical, Version 1.22) using the one-set-of-site fitting model. Non-linear least square fitting using one binding site model was used to calculate the association constant ($K_a$). Dissociation constants ($K_d$) were calculated according to the formula $K_d = 1/K_a$.

## Electrophoretic mobility shift assay

EMSA experiments were carried out on a 5% (w/v) native polyacrylamide gel in Tris-acetate-EDTA buffer containing 40 mM Tris, 20 mM acetic acid, and 1 mM EDTA. Purified Sec3 (aa75–260), WT, or M7 mutant Sso2 (aa1–270), and their mixtures (0.5–1 mg/ml, 10 µl) in the presence of 10% (v/v) glycerol were loaded into separate lanes on the native gel. The gel was run at 150 V for 1.5 hr at 4°C, and then stained in an ethanol solution containing 0.025% (w/v) Coomassie brilliant blue G-250 to visualize the protein bands. The same set of samples were also separately checked on a 15% (w/v) SDS-PAGE gel to confirm the presence of target proteins in each loaded sample.

## Size exclusion chromatography

One-hundred µl purified Sec3 PH domain (100 µM) were mixed with equal volume of either WT or M7 mutant Sso2 (aa1–270, 50 µM) in a sample buffer containing 20 mM HEPES (pH 7.5), 100 mM NaCl, and 1 mM DTT. After incubating at 4°C for 30 min, the mixtures were separately run on a Superdex 200 Increase 10/300 GL column (Sigama-Aldrich) with a flow rate of 0.5 ml/min. Samples from each elution peak were checked on an SDS-PAGE gel to visualize protein content in that peak.

## Construction of strains and plasmids for in vivo assays

All bacteria and yeast strains used in our in vivo studies are listed in **Supplementary files 1 and 2**. To generate the various *sso2* mutants, a fragment of the *SSO2* gene consisting of the promotor region (242 bp) and a C-terminally truncated ORF (1–659 bp) was amplified by PCR. The PCR product was digested and ligated into *Kpn*I/*Hind*III enzyme sites in an integrative pRS306-based vector. The *sso2* mutations were generated by QuikChange Lightning Site-Directed Mutagenesis Kit (Agilent, #210518).

Plasmids carrying the mutant alleles were linearized by digestion with the *Msc*I enzyme to promote integration into the *SSO2* locus and introduced into yeast strains (an *sso1⊗* strain for mutants *sso2M1-M6* or wt for *sso2M7*) by the lithium acetate method. Transformants were selected on SC-Ura plates. Multiple independent transformants were grown in SC-Ura medium overnight. To select for Ura- 'loop-out' segregants, cells (100 µl) were collected, washed in sterile water, and then plated on YPD+5-FOA plates. Integration and loop-out events leading to the genomic expression of the *sso2* mutations were verified by sequencing PCR products. Due to genetic instability issues, we used a two-step procedure to construct an *sso1⊗ sso2M7* strain. A wt strain was transformed with plasmid NRB1659 and then Ura- cells were selected by growth on 5-FOA plates. The selected *sso2M7* strain was crossed with an *sso1⊗SSO2* strain, and after dissection haploid *sso1⊗ sso2M7* spores were identified by PCR analysis.

## Growth test

Mutants were grown in yeast extract peptone dextrose (YPD) medium overnight to stationary phase; 0.04 $OD_{600}$ units of cells were resuspended in 200 µl sterile water and diluted in fivefold serial dilutions. Cells were spotted on YPD plates and incubated at 25°C, 34°C, or 37°C for 2 days.

## Bgl2 secretion assay

The Bgl2 secretion assay was carried out as previously described by **Yuan et al., 2017**. Briefly, 15 ml of yeast cells were grown at 25°C in YPD medium overnight to early log phase (~0.5 $OD_{600}$/ml), then shifted to 37°C for 90 min. Six $OD_{600}$ units of cells from each strain were harvested by centrifugation at 900× *g* for 5 min. Cell pellets were resuspended in 1 ml of ice-cold 10 mM $NaN_3$ and 10 mM NaF, and then incubated on ice for 10 min. The cell suspension was transferred to 1.5 ml microfuge tubes, pelleted, and resuspended in 1 ml of fresh prespheroplasting buffer consisted of 100 mM Tris-HCl (pH 9.4), 50 mM β-mercaptoethanol, 10 mM $NaN_3$, and 10 mM NaF, then incubated on ice for 15 min. Cells were then pelleted and washed with 0.5 ml of spheroplast buffer (50 mM $KH_2PO_4$-KOH [pH 7.0], 1.4 M sorbitol, and 10 mM $NaN_3$). Cells were resuspended in 1 ml of spheroplast buffer containing 167 µg/ml zymolyase 100T (Nacasai Tesque) and incubated at 37°C in a water bath for 30 min. Spheroplasts were spun down at 5000× *g* for 10 min, and 100 µl of the supernatant from each tube was transferred into a new 1.5 ml tube and mixed immediately with 34 µl of ×4 SDS sample buffer (the external pool). All of the remaining supernatant was discarded. The pellets (spheroplasts) were resuspended in 100 µl ×1 SDS sample buffer (the internal pool). Samples were boiled for 10 min and proteins were separated

on a 10% SDS-PAGE gel. Bgl2 was visualized by western blotting with anti-Bgl2 rabbit polyclonal antibody at 1:5000 dilution (provided by the laboratory of Randy Schekman, University of California, Berkeley). For quantitation of Bgl2, images of western blots were captured using the ChemiDoc system (Bio-Rad) and multiple images were collected to ensure an unsaturated signal. Serial dilutions of a control sample were run in parallel to establish a standard curve. The electrophoretic bands were quantitated using ImageJ software (https://imagej.nih.gov).

## Invertase secretion assay

The invertase assays were performed as previously described by *Yuan et al., 2017*. Yeast strains were grown at 25°C in YP medium containing 5% (w/v) glucose overnight to early log phase (~0.5 $OD_{600}$/ml). Then, 1 $OD_{600}$ unit of cells was transferred to two sets of 15 ml centrifuge tubes and pelleted at 3500 rpm for 5 min. The first set of cells were washed in 0.5 ml sterile water and resuspended in 1 ml ice-cold 10 mM $NaN_3$, then kept on ice as 0 min samples. The second set of cells were washed once in sterile water and resuspended in 1 ml YP+0.1% glucose medium, then incubated at 37°C in a water bath with shaking for 45 min. Cells were pelleted and resuspended in 1 ml 10 mM $NaN_3$ as 45 min samples; 0.5 ml cells from each 0 or 45 min samples were used to measure the external invertase. Another 0.5 ml cells of 0 or 45 min samples were resuspended in spheroblast buffer containing 50 µg/ml zymolyase and incubated at 37°C in a water bath for 45 min to generate spheroplasts. All internal samples were lysed with 0.5 ml 0.5% Triton X-100. A 20 µl aliquot from each internal and external sample was used for measuring invertase. The percentage of invertase accumulation was calculated by the formula of $(Int_{45m}-Int_{0m})/[(Ext_{45m}-Ext_{0m}) + (Int_{45m}-Int_{0m})]$.

## Electron microscopy

Control (*sso1⊗/SSO2*) or *sso2* mutant cells were grown at 25°C in YPD to an $OD_{600}$ of ~0.5 and then processed for electron microscopy study as previously described (*Yuan et al., 2017*). In brief, ~10 $OD_{600}$ units of cells were collected using a 0.22 µm filter apparatus, washed with 10 ml of 0.1 M cacodylate (pH 6.8), then resuspended in 10 ml of fixative (0.1 M cacodylate, 4% glutaraldehyde, pH 6.8). Cells were fixed at room temperature for 1 hr and then shifted to 4°C for 16 hr. The next day, cells were washed twice with 50 mM KPi (pH 7.5), and then digested in 2 ml of 50 mM KPi buffer containing 0.25 mg/mL Zymolyase 100T at 37°C for 40 min in a water bath with gentle shaking. Cells were then washed twice with ice-cold 0.1 M cacodylate buffer and resuspended in 1.5 ml of cold 2% (w/v) $OsO_4$ in 0.1 M cacodylate buffer. Cells were incubated for 1 hr on ice, washed three times with water, and then incubated in 1.5 ml of 2% (w/v) uranyl acetate at room temperature for 1 hr. Cells were dehydrated by a series of ethanol washes and incubated overnight in Durcupan resin. Cells were embedded in fresh Durcupan resin and baked at 60°C for at least 48 hr. Sections were stained with lead citrate and uranyl acetate, and images were acquired using a transmission electron microscope (Tecnai G2 Spirit; FEI) equipped with a CCD camera (UltraScan 4000; Eagle).

## Fluorescence microscopy and quantitative localization analysis

### GFP-Snc1 localization

Yeast strains expressing GFP-Snc1 were grown at 25°C to early log phase $OD_{600}$ 0.3–0.6 in SC-Ura medium. Five-hundred µl of cells were harvested and resuspended in growth medium for fluorescence imaging. Cultures were shifted to 37°C for 90 min, then 500 µl samples were collected for fluorescence imaging. Images were acquired with a ×100 oil-immersion objective on a widefield fluorescence microscopy system equipped with a Hamamatsu ORCA-ER camera. For each sample, z-stacks with a 200 nm slice distance were generated. Images were analyzed using Volocity software 4.8 (Quorum Technology, Inc). For quantitation studies, three independent transformants from each condition were examined under the same condition and 100–200 cells from each strain were scored. Three separate experiments were used to calculate the SD.

### LatA treatment of Sec4-GFP and Sec3−3×GFP

Yeast strains were transformed with an integration vector expressing GFP-Sec4 or Sec3−3×GFP. Cells were grown in YPD to around $OD_{600}$ 0.5, then 1 $OD_{600}$ unit cells were collected and resuspended in 50 µl SC-Ura medium. One µl of 10 mM LatA was added to the cell suspension and incubated at 25°C for 15 min. Cells were fixed with 3.7% (v/v) formaldehyde for 60 min and washed twice in 0.5 ml

PBS. Cells were imaged on a widefield fluorescence microscope. Quantitation analysis was done as described above. Three separate experiments were used to calculate the SD.

## Co-immunoprecipitation

Strains were grown at 25°C overnight to an $OD_{600}$ around 1.0. Seventy $OD_{600}$ units of cells were collected from each strain. Cell lysates were prepared as described previously with modifications (*Yue et al., 2017*; *Liu et al., 2022*). Briefly, cell pellets were washed once in cold water and resuspended in ice-cold lysis buffer (50 mM Tris-HCl, pH 7.5, 100 mM NaCl, 1 mM EDTA, 5 mM NaF, 1 mM sodium pyrophosphate, and 1 mM DTT) and a protease inhibitor cocktail (Roche). Cells suspensions were transferred to 2 ml screw cap tubes containing prewashed 2 mg zirconia/silica 0.5 mm beads. Cell were lysed using a bead beater; 1% (v/v) Triton X-100 was added to the cell lysates and incubated for 15 min at 4°C. The cell lysates were then spun at 20,000× *g* for 30 min and supernatants were incubated with 10 µl prewashed anti-Flag agarose beads (Sigma, A2220) at 4°C for 3 hr. Beads were washed five times with lysis buffer containing 0.1% (v/v) Triton X-100. Proteins bound on the beads were eluted with ×1 sample buffer. Proteins were detected with anti-Flag antibody (Sigma, F1804, monoclonal, 1:1000) or anti-Sso antibody (rabbit antiserum, 1:2000).

Coordinates and structure factors for the Sec3-Sso2 complex have been deposited in the Protein Data Bank with accession code 7Q83 (https://www.rcsb.org/structure/7Q83).

## Acknowledgements

We thank the staff at the beamline of ID23-1 at the European Synchrotron Radiation Facility (ESRF) for their help with X-ray diffraction. This work was supported by the grants P28231-B28 and I4960-B from the Austrian Science Fund (FWF) to GD and GM35370 from the National Institutes of Health to PN.

## Additional information

### Funding

| Funder | Grant reference number | Author |
| --- | --- | --- |
| Austrian Science Fund | P28231-B28 | Gang Dong |
| Austrian Science Fund | I4960-B | Gang Dong |
| National Institutes of Health | GM35370 | Peter Novick |
| European Synchrotron Radiation Facility | Austrian Crystallographic Diffraction Consortium | Gang Dong |

The funders had no role in study design, data collection and interpretation, or the decision to submit the work for publication.

### Author contributions

Maximilian Peer, Validation, Investigation, Visualization, Methodology; Hua Yuan, Software, Formal analysis, Validation, Investigation, Visualization, Methodology, Writing - original draft; Yubo Zhang, Investigation; Katharina Korbula, Formal analysis, Validation, Investigation, Visualization, Methodology; Peter Novick, Data curation, Supervision, Funding acquisition, Writing - original draft, Project administration, Writing - review and editing; Gang Dong, Conceptualization, Resources, Data curation, Software, Formal analysis, Supervision, Funding acquisition, Validation, Investigation, Visualization, Methodology, Writing - original draft, Project administration, Writing - review and editing

### Author ORCIDs

Maximilian Peer  http://orcid.org/0000-0001-5032-8029
Peter Novick  http://orcid.org/0000-0003-0401-7857
Gang Dong  http://orcid.org/0000-0001-9745-8103

### Decision letter and Author response

Decision letter https://doi.org/10.7554/eLife.82041.sa1

Author response https://doi.org/10.7554/eLife.82041.sa2

## Additional files

### Supplementary files
- MDAR checklist
- Supplementary file 1. Bacteria strains.
- Supplementary file 2. Yeast strains.
- Source data 1. Original SDS and native protein gels.

### Data availability
Diffraction data have been deposited in PDB under the accession code 7Q83.

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
