## [Editor Report]

This paper describes a mechanistic and structural investigation of a crucial step in exocytosis that involves the assembly of two t-SNAREs with a v-SNARE. The authors solve the structure of Sec3, a component of the exocyst vesicle tethering complex and the t-SNARE Sso2, and identify a previously unknown binding site that helps to explain the mechanism of this rate-limiting step. The results are of great interest to scientists studying exocytosis, membrane fusion, and protein-protein interaction.

---

## [Decision Letter]

[Editors' note: this paper was reviewed by Review Commons.]

---

## [Author Response]

General Statements

We would like to thank both reviewers for their thorough and constructive evaluation and comments on our manuscript. Following their suggestions, we have reworked our manuscript and added several pieces of new data to address questions from them, including (1) evaluation of how M7 mutant of Sso2 affects its interaction with Sec3 using three independent methods (in vitro); (2) investigation of how the M7 mutant affects the interaction of Sso2 with Sec3 by co-immunoprecipitation (in vivo). We hope that, with all these further introduced changes, this manuscript will be suitable for publication in your journal. Detailed point-to-point responses are shown below.

Point-by-point description of the revisions

Reviewer #1 (Evidence, reproducibility and clarity (Required)):Using the entire cytoplasmic domain of Sso2 and protein crystallization, Peer and colleagues show that two N-terminal peptides (NPY) of Sso2 synergistically interact with the Sec3 PH domain. This interaction provides an additional low affinity binding site to the previously published interface between the Sso2 four-helix bundle and the PH domain. Mutagenesis, in particular of both NPY motifs, results in reduced cell growth, in the accumulation of transport vesicles at the budding site, and in decreased secretion of invertase and Bgl2. The paper is well written, the data are convincing and the characterization of these novel peptide interaction sites clearly advances the field. Although, the exact role of the Sec3 NPY – Sec3 interaction still needs to be established, the overall functional relevance is apparent and thus the paper could be published with minor changes.

We really appreciate the reviewer for his/her positive comments and clear/constructive feedbacks.

Nevertheless, the authors may consider to address the following issues to improve the manuscript.To strictly exclude the possibility that the Sso2 NPY motif also interacts with other components of the exocytosis machinery (e.g. Sec1), thereby causing the observed phenotypes, Sec3 mutagenesis of the NPY motif-binding site would be required.

It would be a good idea to generate reverse mutants on Sec3. However, the pocket on Sec3 bound by the NPY motifs of Sso2 is mostly hydrophobic and contains many semi-buried residues that are in close contact with other residues in the hydrophobic core of structure (including L78, Y82, I109, V112, V208, etc.; see **Figure 2-figure supplement 2** D and E) and thus essential in maintaining the folding of Sec3. Making mutations on these residues would destabilize the folding of Sec3. This was why we have not done this as suggested by the reviewer.

The authors suggest that the NPY-peptide binding contributes to the initial interaction/recruitment of Sso2 to the exocytosis site, defined by the localization of Sec3 (exocyst). Further data sustaining this concept/hypothesis could improve the impact of the manuscript. Thus, an experiment analyzing the co-distribution of the Sec3 with Sso2 would directly support the authors' conclusion. (In Figure 7, the authors already show the highly polarized distribution of Sec3-3xGFP.) The M7 mutant could impact the distribution of Sso2. In addition, it would be helpful to determine to which degree the Sso2 NPY – Sec3 PH domain interaction increases the overall affinity of Sso2 for the Sec3 PH domain; e.g. comparison of the binding of Sso2 (1-270) wt and M7 to Sec3 PH domain using ITC.

We greatly value the reviewer’s suggestion. For the suggestion to investigate how the M7 mutant affects the co-distribution of Sso2 with Sec3 in yeast, we have tried a variety of conditions with both the original serum and affinity purified Sso antibodies. In neither case did we see a clear concentration at sites where we would expect to see Sec3, such as the tips of small buds. We were able to see some detectable concentration of HA-tagged Sso2 in small buds using antiHA Ab, but it would be difficult to tag the M7 mutant at the same site since it is so close to the M7 mutation. We are also worried that the tag might interfere with Sec3 binding due to the proximity. Given the lack of detectable concentration of WT Sso2, it would not be possible to see a loss of localization in M7.

For the suggestion to check the binding of Sec3 with either the WT or M7 mutant of Sso2 (aa1-270), we have generated M7 mutant within the same fragment of Sso2 as the WT (i.e. aa1-270) and carefully checked how this M7 mutant affects the interaction of Sso2 with the Sec3 PH domain using three independent methods. Our ITC data show that WT Sso2 bound Sec3 very robustly, with a Kd of approximately 2 µM (Figure 8C). Surprisingly, however, the M7 mutant of Sso2 (aa1-270) completely abolished its interaction with Sec3 (Figure 8D). To further verify this observation, we carried out electrophoresis mobility shift assays (EMSA) and size-exclusion chromatography (SEC). Our EMSA data on a native PAGE gel shows that WT Sso2 (aa1-270) bound Sec3, whereas the M7 mutant did not (**Figure 8-figure supplement 1** A and B). Similarly, our SEC data demonstrate that Sec3 was coeluted with WT Sso2 in the higher molecular weight peak; in contrast, Sec3 and the M7 mutant of Sso2 (aa1-270) were eluted in separate peaks and no stable complex of the two was formed (**Figure 8-figure supplement 1** C and D). All these new data confirm that the NPY motifs play an essential role in maintaining the stable interaction between Sso2 and Sec3, which would explain why the M7 mutant gave such dramatic phenotype in vivo (Figure 4 B-E; Figure 5 D-F; Figure 6 D and E).

Minor point: In the discussion, the authors should mention to which degree the NPY binding site within Sec3 is accessible for / occupied by other known exocyst components, or PI(4,5)P2, etc.

Thank you for the suggestion. A new diagram has been added to Figure 9E to compare the structures of the previously reported Sec3/Rho1 complex and the Sso2/Sec3 complex determined by us. It shows that the NPY binding site on Sec3 is on the opposite side of the membrane-binding surface patch. The NPY binding site is also far away from the Rho1 interacting site on Sec3 and thus does not interfere with Rho1 binding either.

Reviewer #1 (Significance (Required)):The manuscript significantly contributes to our understanding of how the vesicle tethering machinery interacts and coordinates the assembly of the membrane fusion machinery and will be of broad interest in the field of membrane trafficking. I am not an expert in X-ray crystallography.

We sincerely appreciate this reviewer’s positive feedbacks.

Referees cross-commentingI agree with the comments of the other reviewer. It would be nice to show the effect of the M7 mutant in a reconstituted liposome fusion assay, but as already mentioned this may require an additional collaboration. Whether the relatively weak Sec3 – NPY interaction can be resolved in the liposome fusion assay needs to be shown.

Please check our detailed answer to the other reviewer’s question about this.

Reviewer #2 (Evidence, reproducibility and clarity (Required)):The manuscript of Peer et al. Describe the structural characterization of the interaction of the syntaxin-like Sso2 protein with the exocyst subunit Sec3. The authors identify here a dual NPY motif at the N-terminal part of Sso2 that binds to Sec3 and thus confers functionality. Using x-ray crystallography, they show a nearly full-length Sso2 in complex with Sec3, which reveals how Sso2 binds to Sec3. Subsequent mutagenesis shows that both NPY motifs act together in binding, and are both required for functionality in vivo, using established assays in localization of exocyst subunits, secretion assays and growth tests. Their data suggest an overall model how Sso2 is efficiently recruited by exocyst to promote vesicle secretion.This is an overall very complete and clear manuscript, where the authors nicely demonstrate, how the two NPY motifs both contribute to efficient Sso2 interaction with Sec3. Their data further show that each motif alone can contribute to function, whereas loss of both motifs (the M7 mutant) result in deficient binding. Likewise, their established assays to determine cellular importance of the NPY motifs in Sso2 show that trafficking and localization in the secretory pathway is strongly impaired in the mutant. I only have a few questions and suggestions.

Thank you for the positive feedback.

1. The authors present in Figure 4 the mutants. I recommend to show the alignment of the mutants (M5,M6,M7) similar to panel A in Figure S4 here to orient the reader. They could also be listed in Figure 3, since the authors have here the sequences.

Alignment of M5-M7 has been added in Figure 4A as suggested. Thank you.

2. The authors previously showed that Sso2 mutants affect the Sec3 driven assembly and also the fusion. I am wondering if they have the tools ready to also conduct this assay with their M7 mutant, which has the strongest defect. I am aware that this may be challenging if the tools are not established here as in the previous collaboration (Yue et al., 2017). It may provide additional information on the functional crosstalk.

Thank you for the suggestion. However, we do not think it is necessary to perform such assay based on our new results. As shown in Figure 8 C and D and Figure S5, we found that the M7 mutant of Sso2 (aa1-270) completely abolished its interaction with Sec3, which is in contrast to the robust interaction between the WT Sso2 (aa1-270) and Sec3. Therefore, we expect that the M7 mutant would fail to accelerate liposome fusion in the same way as we had previously seen for the WT Sso2.

On the other hand, we have to admit that to perform such assay would indeed be challenging for us as the PhD student who had carried out the in vitro liposome fusion assay has left our previous collaborator’s lab and it would take quite a while to re-establish the assay in our own group and to optimize various parameters in that assay.

3. Along the same line, it would be good if the authors show that the mutation also impairs the interaction of Sec3 and Sso2 in vivo.

We appreciate the reviewer’s suggestion and have carried out coimmunoprecipitation of Sec3-3×Flag and Sso2 from yeast extract to find out how the M7 mutant affects Sso2’s interaction with Sec3 (Figure S6). Our results show that in contrast to the clear signal of WT Sso2 pulled down by Sec3-3×Flag, the pull-down band for the M7 mutant was much weaker and at a similar level to the negative control. This is consistent with what we saw in our in vitro binding assays (Figure 8D; **Figure 8-figure supplement 1**).

4. I really like the similarity of the different Munc18-Syntaxin interactions and the Sec3Sso2 interaction. Do the authors think that Sec3 is an ancestral fragment of a Sec1 like protein, which just maintained this interaction?

This is a very interesting idea. However, it seems too speculative to us to draw such conclusion. It could also be due to co-evolution in function for Sec3 to use a simpler structure (i.e. PH domain) to mimic syntaxin binding of SM proteins and to employ the extra “add-on” NPY motifs as a handle to facilitate and regulate their interaction.

5. Small mistake in the discussion Responses: "plasmas membrane"

This has been corrected. Thank you.

Reviewer #2 (Significance (Required)):Important advance in our understanding of Exocyst function, which deserves publication. I only had minor issues that can be addressed quickly.

We sincerely appreciate the reviewer’s positive feedbacks and constructive suggestions.